# FGFR-2 and Epithelial–Mesenchymal Transition in Endometrial Cancer

**DOI:** 10.3390/jcm11185416

**Published:** 2022-09-15

**Authors:** Olga Adamczyk-Gruszka, Agata Horecka-Lewitowicz, Jakub Gruszka, Monika Wawszczak-Kasza, Agnieszka Strzelecka, Piotr Lewitowicz

**Affiliations:** 1Department of Gynecology and Obstetrics, Collegium Medicum, Jan Kochanowski University, 25-317 Kielce, Poland; 2Department of Obstetrics and Gynecology, Province Hospital, 25-736 Kielce, Poland; 3Institute of Health Sciences, Collegium Medicum Jan Kochanowski University, 25-317 Kielce, Poland; 42nd Department of Obstetrics and Gynecology, Medical University of Warsaw, 02-091 Warsaw, Poland; 5Department of Surgical Medicine with the Laboratory of Medical Genetics, Institute of Medical Sciences, Jan Kochanowski University, 25-317 Kielce, Poland; 6Department of Clinical and Experimental Pathology, Institute of Medical Sciences, Jan Kochanowski University, 25-317 Kielce, Poland

**Keywords:** endometrial cancer, FGFR-2, β-catenin, EMT, next-generation sequencing

## Abstract

Background. At present, EC staging is based on the WHO conservative criteria, which only consider the percentage of gland formation. The molecular subgrouping of EC recently proposed by the Cancer Genome Atlas (TCGA) represents a milestone in precise molecular-based patient triage. The present study aimed to investigate the influence of FGFR-2 on the epithelial–mesenchymal transition (EMT) and whether it can lead to endometrial cancer dedifferentiation. Methods. One hundred and three White female patients with confirmed EC were enrolled in our research. For the analysis, we performed next-generation sequencing and immunohistochemical analyses of E-cadherin, β-catenin, and vimentin. Results. Tumor grade progression was closely correlated with LVI (*p* = 0.0338), expression of vimentin (*p* = 0.000), tumor budding (*p* = 0.000), and lack of E-cadherin (*p* = 0.0028). Similar observations were noted with regard to TNM/FIGO stage progression. In terms of FGFR-2 mutation, we found the following correlation *p*-values: LVI (*p* = 0.069), expression of vimentin (*p* = 0.000), tumor budding (*p* = 0.000), and lack of E-cadherin (*p* = 0.000), RFS (*p* = 0.032), ECSS (*p* = 0.047). Conclusions. FGFR-2 is the important factor influencing on EMT.

## 1. Introduction

In recent decades, the morbidity and mortality associated with endometrial cancer (EC) have been increasing despite diagnostic progress. For decades it was believed that obesity and longtime estrogen exposure were particularly associated with EC. Moreover, this malignancy is much more common in high-income countries due to their high rates of obesity [1]. To date, the oncological grading system for EC is based on WHO rules where solid areas and gland formation make up its core [2]. The Cancer Genome Atlas (TCGA) has recently proposed molecular subgroupings that represent a milestone in precise, molecular-based patient triage [3]. Mechanically, a new diagnostic algorithm has been developed that gives us more insights into EC’s molecular pathways, tumor histology, and biological behavior. In daily practice the use of such molecular subgrouping would be cumbersome, but advances in molecular techniques have made these tests cheaply available. For integrated ‘histo-molecular’ diagnosis, pathology reports should include information about the mutational status of the POLE gene and the results of the immunohistochemical assays of p53, MSH6, and PMS2 [3,4].

The association between hormone receptors and prognostic variables (FIGO stage, grade, and survival) has been well documented in EC [5]. A recent study reports the involvement of β-catenin in tumor progress via the impact of epidermal growth factor receptor rearrangements. Junctional adhesive molecule-A (JAM-A) can regulate EGFR levels by modulating β-catenin localization [6]. The environment where cancer grows is composed of stromal cells and inflammatory cells and their outputs in the form of growth hormones, ligands, or cytokines. It promotes epithelial cell proliferation and cell differentiation as well. The last decade has provided us with insights into transcription factors that can modulate E-cadherin expression and induce deep changes in the cytoskeleton allowing for both its infiltration by singular cells and wider invasions. These factors include the transforming growth factor-beta, epidermal growth factor, insulin growth factor 1, interleukin, vascular endothelial growth factor, platelet-derived growth factor, integrin/integrin-linked kinase, Notch, fibroblast growth factor, and Wnt/b-catenin signaling pathways. Most of these signals exert their action on E-cadherin repression through the modulation of a set of pleiotropically acting transcription factors, including members of the Snail (Snail and Slug) and basic helix–loop–helix (E47 and Twist) families, as well as two double zinc finger and homeodomain (Zeb1 and Zeb2) factors [7,8,9,10]. Moreover, stromal desmoplasia provides cancer-associated fibroblasts that in turn induce the secretion of EGF, TGF-β, HGF, and FGF-2 [11].

Estrogen-induced endometrial stromal cells could secrete fibroblast growth factor (FGF), which in turn binds to the fibroblast growth factor receptor family (FGFR) receptors on the cell membrane and acts as a pro-proliferative factor. Moreover, FGFR-2 receptor mutation leads to its overexpression and overactivity. A first large-cohort study concerning FGFR2 in EC started over decade ago. Results unveiled 10% FGFR2 mutation contribution in endometrioid EC. Moreover, the authors noted shorter RFS and OS in cases harboring FGFR2 mutation [12,13]. Another study has demonstrated the cancer microenvironment’s impact to cell adhesion, biological adhesion, bone development, and extracellular matrix organization. Interestingly, the authors observed impacts on the Wnt pathway and β-catenin via protocadherin α gene cluster (PCDHA) disturbances, which play a crucial role in cadherin maintenance [14,15].

Aim. We intend to estimate the contribution of FGFR2 to the epithelial–mesenchymal transition (EMT), tumor budding, and prognosis.

## 2. Materials and Methods

A total of one hundred and three White female patients with confirmed EC were enrolled. All patients underwent surgery and other diagnostic oncological procedures between 2005 and 2017. The collective evaluation data and follow-up data were tabulated. To align tumor staging, each case was re-diagnosed according to the Eighth Edition of the TNM Classification system and the recent ESMO Clinical Practice Guidelines for diagnosis, treatment, and follow-up [16,17].

All participants underwent surgical treatment without previous radio-chemotherapy to conduct a credible comparative analysis of the tumor characteristics, treatment, and unchanged molecular profiling. A complete characteristic of studied group was described in our previous publication [18].

To achieve the goal, we used an Illumina Hot Spot Cancer Panel (Illumina Inc., San Diego, CA, USA). The gene panel comprised BL1, EGFR, GNAS, KRAS, PTPN11, AKT1, ERBB2, GNAQ, MET, RB1, ALK, ERBB4, HNF1A, MLH1, RET, APC, EZH2, HRAS, MPL, SMAD4, ATM, FBXW7, IDH1, NOTCH1, SMARCB1, BRAF, FGFR1, JAK2, NPM1, SMO, CDH1, FGFR2, JAK3, NRAS, SRC, CDKN2A, FGFR3, IDH2, PDGFRA, STK11, CSF1R, FLT3, KDR, PIK3CA, TP53, CTNNB1, GNA11, KIT, PTEN, and VHL. The sequencing target was 2800 COSMIC mutations from 50 oncogenes and tumor suppressor genes.

In addition, we performed a tissue microarray (TMA Master II, 3DHistech) for the immunohistochemistry analysis. We performed the immunohistochemical assays using the automated IHC/ISH slide staining system BenchMark Ultra (Ventana Medical Systems; Roche Group, Tucson, AZ, USA). After the deparaffinization and rehydration of the samples, we performed the unmasking processes using a CC1 (Ventana Medical Systems; Roche Group, Tucson, USA) and incubation with primary antibodies (the time and temperature of both antigen retrieval and primary antibody incubation followed the manufacturer’s recommendations). In addition, further routine steps were performed. Moreover, we used the Ventana ultra-View Universal DAB Detection Kit and Opti View Detection Kit. The immunohistochemistry details are presented in Table 1.

As the tumor cell budding phenomenon has been described for colorectal cancer, we decided to explore it in endometrial cancer. Tumor budding was performed according to the method of Ueno et al. and the recent consensus of the International Tumor Budding Consensus Conference [19,20]. We decided to confirm the EMT in cases that were vimentin positive, E-cadherin negative, and had four or more tumor buds per 0.785 mm^2^. Substantial Lymphovascular invasion (LVI) was defined as ≥4 LVSI-positive vessels in at least one H&E slide.

### 2.1. Molecular Analysis

DNA isolation: Cancer genomic DNA was extracted from formalin-fixed paraffin-embedded tissue using a MagCore^®^ Genomic DNA FFPE One-Step Kit (RBC Bioscience, Taiwan). The quality was quantified using a DeNovix DS-11 Spectrophotometer (DeNovix, Wilmington, DE, USA) and a QuantiFluo^®^ ONE dsDNA System (Promega, Madison, WI, USA).

The assay generates a library of 207 gene-specific amplicons and targets ~2800 clinically relevant mutations.

Sequencing: The products were analyzed via next-generation sequencing (NGS) using an Illumina platform, MiSeq Dx.

Data analysis: the analysis of NGS data was performed using the GALAXY platform (usegal-axy.org, accessed on). Sequencing reads (FASTQ files) were aligned to the human reference genome hg19 using the Bowtie2 tool. Variant calling was performed using the Varscan2 tool. The parameters used for the analysis were minimum allele frequency—0.05, minimum quality—20, and minimum coverage ×80. All variants were annotated with ANNOVAR (https://wannovar.wglab.org, accessed on 2 May 2022). The results were visualized using the R Bioconductor package by Maftools (http://bioconductor.org/, accessed on 2 May 2022). More details have been described in previous study [18].

### 2.2. Ethical Statement

This retrospective study used human tissues for the experiment was performed in accordance with the updated ethical standards of the Declaration of Helsinki (2004). In addition, the study was approved by the Ethical Commission of the Faculty of Medicine and Health Science, Jan Kochanowski University, Kielce, Poland on 3 December 2019 (decision No. 47/2019).

### 2.3. Statistical Analysis

Descriptive statistics were gathered in order to summarize the data in a manageable form. Quantitative data are reported as mean, standard deviation, median, and range. Categorical data are expressed as number and percentage distributions. The chi-square test or Fisher’s exact test was applied to compare proportions, and a multivariable logistic regression model was used to assess the relationship between the targeted genes. The follow-up period was calculated as the number of years from the date of surgery to disease recurrence. Deaths that occurred from causes other than cancer were recorded. The last contact with the patient is also presented. The univariate associations between disease-free survival in selected patients and tumor characteristics were evaluated using the univariate Cox proportional hazard model. Analyses of continuous variables were dichotomized in the median. To identify the independent prognostic factor for disease-free survival, multivariate Cox proportional hazard models with backward selection (with the cut-off at 0.05) were performed on variables that were statistically significant in univariate and multivariate analyses.

All statistical tests were two-sided, and values <0.05 were considered significant. The computations were performed using STATISTICA (data analysis software system) version 12 (2014) (StatSoft, Inc., Tulsa, OK, USA, http://www.statsoft.com/, accessed on 2 May 2022).

## 3. Results

### 3.1. Epithelial–Mesenchymal Transition Analysis

Table 2 presents the crucial data influencing the prognosis with regard to vimentin, EMT, E-cadherin, Β-catenin, and estrogen receptors. As can be seen, vimentin positive/E-cadherin negative cells and EMT strongly correlated with stage and grade progress.

Tumor grade progression was closely correlated with LVI (R = 0.2113, *p* = 0.0338), expression of vimentin (R = 0.5344, *p* = 0.000), tumor budding (r = 0.4867, *p* = 0.000), Β-catenin (R = 0.410, *p* = 0.044), and lack of E-cadherin (R = 0.2950, *p* = 0.0028). Similar observations were noted with regard to TNM/FIGO stage progression—LVI (R = 0.6949, *p* = 0.000), expression of vimentin (R = 0.2573, *p* = 0.009), tumor budding (R = 0.3098, *p* = 0.000), Β-catenin (R = 0.437, *p* = 0.032), and lack of E-cadherin (R = 0.3291, *p* = 0.000). Moreover, mutation of FGFR-2 contributed to E-cadherin expression decline (*p* < 0.001) and EMT (*p* < 0.001).

### 3.2. Epithelial–Mesenchymal Transition and FGFR-2 Mutation Impact on Survive

As previously reported the FGFR-2 mutation was observed in 14% [18]. The comparative analysis of FGFR-2 mutation vs grade, stage, survive, age, and BMI are presented in Table 3. FGFR2 mutation occurred very rare in FIGO G1 tumors (*p* = 0.016). Analysis of low vs high grade FIGO stage pointed FGFR2 as more common in advanced tumors (*p* = 0.034), and finally the shorter time of RFS and ACSS was observed in cases with FGFR2 mutation (*p* < 0.05).

The OS correlation was positive for beta catenin (test log-rank for beta-catenin—*p* = 0.03977; HR test 0.456, *p* = 0.041). The Kaplan–Meier curves in Figure 1 present the strongest impacts were caused by LVI+, E-cadherin loss, and tumor budding. The Cox model (Table 4) provided us similar information but multivariant test pointed LVI as the most important one.

As can be seen in Table 4, LVI had the strongest impact on OS. The following question was: is there a correlation between LVI and targeted genes panel?

The mutational contributions were as follows: PTEN 49%, PIK3CA 35%, KRAS 25%, TP53 20%, FGFR-2 14%, CTNNB1 12%, FBXW7 9%, ATM 1%, ALK1 1%, and APC 1%. An attempt to join the EMT’s features with high-grade mutations showed the following results: TP53, KRAS vs. vimentin, E-cadherin, and tumor budding (*p* > 0.05). Lympho-vascular invasion was more commonly observed in TP53 mutated tumors (R = 0.3138, *p* = 0.009). The opposite results were obtained with respect to FGFR-2 mutation. Here, were noted the results as follows: LVI (R = 0.2181, *p* = 0.069), expression of vimentin (R = 0.8199, *p* = 0.000), tumor budding (R = 0.6407, *p* = 0.000), and lack of E-cadherin (R = 0.6948, *p* = 0.000). Figure 2 depicts vimentin reactivity in poorly differentiated cancer and change in cancer cell morphology called EMT.

To analyze the interaction between FGFR-2 and β-catenin, we excluded all cases with CTNNB-1 and APC mutations. This allowed us to analyze other ways of Wnt pathway activation. Beta-catenin showed a correlation with FGFR-2 mutation (R = 0.2508, *p* = 0.04058). Moreover, beta-catenin expression was observed in the progress stage (R = 0.2209, *p* = 0.0263) with a prognostic impact on RFS (R = 0.2049, *p* = 0.0388). Worse clinical outcomes were observed in patients with a BMI between 30 and 34.99 with beta-catenin expression (R = 0.1931, *p* = 0.0410).

## 4. Discussion

A remarkable change in the molecular subgrouping of EC has been witnessed in recent years. Low- and high-grade EC groupings have been proposed, where the latter is driven by TP53 mutation, high copy number variations, and microsatellite stability. This study attempted to discuss the internal crossing pathway involved in EC progress, especially in the context of the EMT. Ueno et al. first proposed their criteria for colorectal cancer grading to the WHO almost 20 years ago [19]. In our study, we observed the same cancer cell attributes, including the loss of intercellular connections before becoming a hybrid epithelial–mesenchymal cell. However, it is less common then in colorectal cancer, poorly differentiated EC present the same EMT attributes. The available data concerning FGFR-2 and β-catenin are extremely poor. Their cooperation was reported in embryogenesis, wound healing, and even cancer [20,21,22].

A recent study indicated that FGFR-2 mutation is a causative factor of the EMT in EC [23]. The dysregulation of FGF secretion and FGFR expression in stromal, estrogen positive cells, and also cancer cells can be a driving force in cancer progression [24]. Our results on aspects of the EMT confirmed a direct negative impact on OS. Interestingly, a distinctive tumor attribute of lympho-vascular invasion (LVI) was observed more frequently in high-grade TP53 mutated tumors, especially with EMT. Our results unanimously confirm that EMT features are useful for prediction but only LVI reached the predictive validity in the multivariant Cox model. The approach to explain LVI mechanism led us to TP53 pathway, but the FGFR2 mutation in this field is still unclear. Our results confirmed previous data concerning RFS and OS [13]. We observed shorter time of RFS and ACSS with no final impact on OS.

Much is known about E-cadherin, Snail1/2, vimentin, and other attributes of the EMT, but the role of catenin is still unclear in this field. For many years, β-catenin has been perceived as a factor that worsens prognosis in EC. Recently, a new paper put a new light on the β-catenin-dependent EMT. In essence, Wnt/β-catenin pathway activation plays a pivotal role in the epithelial–mesenchymal transition (EMT). The Wnt pathway is closely associated with the overexpression of Snail, Slug, and Twist, which control the downregulation of the adherens junction epithelial protein (E)-cadherin and the upregulation of the mesenchymal-specific marker neuronal(N)-cadherin during the EMT, promoting cell migration [25]. A recent paper by Ferguson et al. perfectly explains the action and regulation of the FGF axis. In the context of β-catenin, the authors first joined FGFR-2 with E-cadherin and indirectly with β-catenin. FGFR interacts directly with cadherins (and NCAMs) through the conserved acid-box region in IgIII [26,27,28,29] and then activates many pathways. In our study, β-catenin was observed in nuclear compartment or focal loss of membrane expression in APC, CTNNB-1, and CDH-1 wild-type tumors. Moreover, these tumors presented FGFR-2 mutation. The results concerning its impact on OS, grade, and stage were positive, but this may be an additional result of FGFR activation. In our study, we found that the loss of tumor cells’ integrality by tumor budding together with vimentin expression vividly promotes the EMT. We observed a tight correlation between the EMT and advanced grade and stage and with general OS. The present study indirectly confirms the cross-talks between transcriptional factors and abnormal gene function as was published by us recently [30].

The role of cancer stem cells (CSC) has been discussed many times in the literature. Their contribution to tumor dedifferentiation and impact on ECM, EMT, and chemoresistance have led to many trials directed towards CSC inhibition [31]. A recent paper by Chen et al. presents the clinical implications of EMT regulation. The presented in vitro and in vivo preclinical studies both demonstrated that isoliquiritigenin (ISL) efficiently suppressed endometrial cancer cell migration and reduced the HEC-1A-LUC tumor metastasis in nude mice by inhibiting the TGF-β/Smad signaling pathway. These findings will likely lead to further research to highlight ISL’s potential in the treatment of endometrial cancer metastasis [31].

## 5. Conclusions

Our study confirmed that EMT would be a reliable biomarker for the prediction of EC outcomes.

FGFR-2 mutation could contribute to EMT and indirectly worsens prognosis.

## Figures and Tables

**Figure 1 jcm-11-05416-f001:**
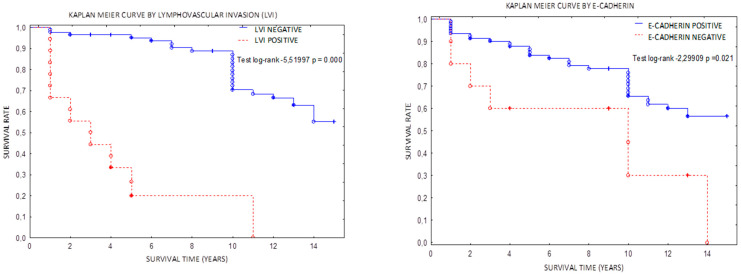
Kaplan–Meier curves for lymphovascular invasion, loss of E-cadherin, β-catenin, and tumor budding.

**Figure 2 jcm-11-05416-f002:**
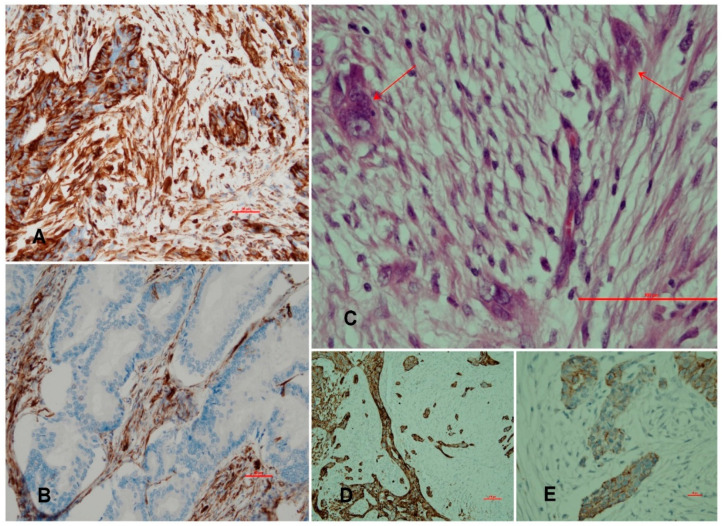
The compilation of five pictures representing of EMT and tumor budding in aspect of FGFR-2 mutation (*p* = 0.000). (**A**) Vimentin+ singular cancer cells forming small groups. (**B**) Vimentin+ stromal cells. Note the lack of expression in well-formed glands. (**C**) H&E staining showing small groups (arrows) of pleomorphic and spindle cells. (**D**) Pancytokeratin staining showing singular and small buds of cancer cells as an example of tumor budding. (**E**) A focal lack of E-cadherin leads to a gradual loss of epithelial characteristics.

**Table 1 jcm-11-05416-t001:** The antibodies’ characteristics.

	Clone	Dilution	Manufacturer	Positive Expression
Β-catenin	14—monoclonal; mouse	1.64 µg/mL; ready to use	Cell Marque	Nuclear/Cytoplasmic
Vimentin	Vim 3B4	ready to use	Ventana	Cytoplasmic
ER	SP1 monoclonal, rabbit	1 µg/mL; ready to use	Ventana	Nuclear
E-cadherin	36 monoclonal; mouse	0.314 µg/mL; ready to use	Ventana	Membranous

**Table 2 jcm-11-05416-t002:** Tumor characteristics and expression of targeting proteins according to grade, stage, and FGFR-2 mutation.

*N = 103*	*G(N)*			*TNM/FIGO*					*Stage Progress*	*Grade Progress*	*FGFR2 Mutation (N)*	*FGFR2 Wild Type (N)*	*p*
	*G1*	*G2*	*G3*	*IA*	*IB*	*II*	*III*	*IV*					
Histopathological type									n/a	n/a			*p* < 0.001
➢ Endometrioid							14	15			10	93	
➢ Non -endometriod	49	39	7	25	21	20	4	5			0	0	
ESMO									n/a	n/a			*p* = 0.021
➢ Low grade	49	17	0	49	20	18	0	0			3	100	
➢ High grade	0	22	7	0	1	2	10	14			7	3	
Substantial lymphovascular invasion (LVI)	0	20	7	0	2	4	7	11	*p* = 0.038	*p* < 0.001	3	24	*p* > 0.05
Vimentin+	0	14	22	0	0	1	16	20	*p* < 0.001	*p* < 0.009	10	2	*p* < 0.001
EMT (tumor budding plus vimentin+, E-cadherin negative)	0	6	14	0	4	5	6	17	*p* < 0.01	*p* < 0.001	10	1	*p* < 0.001
Β-catenin	5	9	1	7	3	5	0	0	*p* = 0.044	*p* = 0.032	6	9	*p* > 0.05
E-cadherin lack	0	1	4	0	0	3	6	10	*p* = 0.002	*p* < 0.00	9	1	*p* < 0.001
ER (77%)	42	33	5	54	10	12	2	1	*p* > 0.5	*p* > 0.5	4	76	*p* < 0.001

**Table 3 jcm-11-05416-t003:** The analysis of FGFR-2 mutation impact on grade, stage and survive.

	FGFR-2 Mutated	FGFR-2 Wild Type	*p*
Age (median)	72	70	*p* = 0.1632
BMI (median)	32.2	34	*p* > 0.05
FIGO G1	2	47	*p* = 0.016
FIGO G2	3	36	*p* > 0.05
FIGO G3	6	1	*p* > 0.05
FIGO stage I/II	3	72	*p* > 0.05
FIGO stage III/IV	7	21	*p* = 0.034
OS (median) (years)	7.7	8	*p* > 0.05
RFS (median) (years)	4	6.2	*p* = 0.032
ECSS (endometrial cancer specific survival; median)	5.2	7.4	*p* = 0.047

**Table 4 jcm-11-05416-t004:** A comparison of univariate and multivariate Cox models.

	Univariate Cox Model				Multivariate Cox Model			
	Chi Square	HR	CI 95%	*p*	Chi Square	HR	*p*	CI 95%
**LVI+**	42.84	0.0779	−1.65 to −0.89	0.000	43.58	0.061	0.000	−3.61 to −1.96
**Tumor budding**	3.997	0.4821	−1.44 to −0.01	0.0455	0.080	0.8338	0.776	−1.43 to 1.07
**Vimentin+**	3.377	0.4934	−0.72 to −0.02	0.066	0.047	0.877	0.827	−1.30 to 1.043
**E−cadherin**	5.238	0.38	−1.79 to −0.13	0.022	0.524	0.628	0.469	−1.72 to 0.79
**FGFR2 mutation**	3.203	0.42	−0.89 to 0.12	0.043	0.2850	1.635	0.59	−1.54 to 0.66

## Data Availability

Data will be available from the corresponding author upon reasonable request.

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
