# Peer review of "FGFR-2 and Epithelial–Mesenchymal Transition in Endometrial Cancer"

_jcm, 2022, doi:10.3390/jcm11185416_

Round 1
Reviewer 1 Report (New Reviewer)
Study by Gruszka O et.al is interesting.
Quantify figures and add statistics for better representation in figure 2.
Can authors comment on the coexpression of FGFR2 and Vimentin? The coexpression graph would better represent the study and support the analysis.
Author Response
Dear Reviewer,
We are truly grateful for your comments. Indeed, a more detailed Figure 2 description was necessary. I can not provide a graphical co-expression of FGFR-2 and vimentin because FGFR-2 was tested by NGS as an aspect of molecular alterations. We mean in the main text it was slightly corrected and should be correct. Kind regards, AuthorsReviewer 2 Report (New Reviewer)
Dear authors,
Thank you for submitting your manuscript investigating the influence of FGFR-2 on the EMT in endometrial cancer.
Positives of the study include – molecular analysis (NGS), long term follow up-
However, the main limitation is the small number of patients included (limitation for prognostic analysis. Furthermore the population is heterogeneous with localized and metastasis patients).
I have several comments/suggestions:
1) Table 2 : Add missing data in the table/%
Problem with the number of FGFR2 mutation in low grade/high grade (sum>103)
2) Number of FGFR2 mutation in the total population? I do not see the information
3) Table 3: number of G1/G2/G3 is different from table 2 (48/45/9 vs 49/39/7)
4) Table 3: OS/RFS/ECSS -> in months or years?
5) Table 4: please add if LVI+/-, vimentin +/-… to better understand the HR. Please add CI 95%
Author Response
Dear Reviewer, We are truly grateful for your comments. Indeed, we made numerical errors that have been corrected. We added FGFR-2 percentage, then we precisely described OS/RFS/ECSS, and finally CI95% . Thank you for your detailed review and valuable comments. Sincerely, AuthorsRound 2
Reviewer 2 Report (New Reviewer)
I think it is important to include number of missing data in table 2
Author Response
Dear Reviewer,
Thank you for your suggestion. Indeed, the information concerning missing data is important. Preparing the study we collected about 180 cases of EC. Only 103 cases provided good quality DNA to NGS. The rest of the cases had significantly fragmented DNA. Today I have only information about technical problems with immunohistochemistry - it was added to Table 2.
Kind regards
Authors
This manuscript is a resubmission of an earlier submission. The following is a list of the peer review reports and author responses from that submission.
Round 1
Reviewer 1 Report
The authors Adamczyk-Gruszka et al. submitted the manuscript entitled “FGFR-2 mutation worsens prognosis of endometrial cancer via epithelial-mesenchymal transition”. It was difficult to understand the real objective of this study. The Title and the Abstract do not correspond to the assays performed. This is a study about the influence of biomarkers on the prognosis of patients with endometrial cancer and there are no experiments in biological models that support the study's conclusions. Therefore, the conclusions/discussion of the authors overstep their data.
Author Response
Dear Reviewer,
I would like to thank you for the detailed review.
On behalf of the Authors
Associate Professor Piotr Lewitowicz MD, PhD
Reviewer 2 Report
I would like to thank for the opportunity to review this manuscript. This manuscript reported commonly mutated genes and the link between EMT, B-Catenin, FGFR2 and tumour budding. The concept and attempt of the authors to link these altered genes and EMT would be interesting. However, this study has several major concerns.
1. Data was presented inadequately and inappropriately. Several clinicopathologic paramaters are not provided. Please include, histologic type (endometrioid vs nonendometrioid endometrial cancer, myometrial invasion, lymphovascular space invasion, ESMO risk (low vs intermediate vs high). It is highly recommended to present Table 4 as the previously published articles e.g
- PMID: 27006490
- DOI: 10.1158/1078-0432.CCR-15-2878
2) Several centers opted molecular classification of endometrial cancer as risk stratification and patient management and this supported by WHO and ESGO as pointed in the introduction. It is highly desired to correlate the EMT and tumour budding with molecular subtypes.
3) The authors should present the univariable and multivariable analyses to show Vimentin, tumour budding, B-catenin the other parameters are independent prognostic factors.
4) The authors should describe the scoring method used for reporting the IHC of vimentin and b-catenin. The authors should report the correlation between B-catenin expression and mutation.
5)The authors should report the Recurrence free survival and disease free survival curves based on the vimentin expression and B-catenin expression. May be low vs high expression for both markers.
6) Please give representative microphotography for the of low and high expression of vementin showing tumour budding.
7) EMT is not adequately defined by vimentin expression neither by tumour budding. It is inadequate data to represent EMT by vimentin expression. The authors should do more markers of EMT, E-cadherin, CDH2, SNAIL1/2, TWIST etc to define EMT. For example loss of E-cadherin and Vimentin could better support the tumour underwent EMT instead of vimentin+ only expression.
8) Previous study reported B-catenin mutation and FGFR2 mutation are matually exclusive and the current data is quite opposite. Could the author give more justification.
9) "vimentin+ cells, EMT and β-catenin correlated with grade and stage progress" this not true. The p-value for stage is >0.5 in Table 4.
10) The conclusion is not supported by the data presented. "Our study confirmed that EMT is a reliable biomarker in the prognosis of EC out-comes. The changes of cell phenotype via FGFR-2 amplification could activate β-catenin via impact to cadherin results in aspect of EMT are valuable for tumor grading". Only vimentin is associated with grade and stage not necessary to represent EMT. There are a number of published papers that link EMT with FGFR2 and the authors should understand FGFR2 has two splice isoforms and only FGFR2c (mesenchymal isoform) splice isoform is a hallmark or reporter for EMT DOI 10.1016/j.molcel.2009.01.025, (DOI 10.1158/1078-0432.CCR-19-4088 . The results in aspect of EMT are valuable for tumor grading" No enough data to support this strong conclusion. The authors should present mechanical study to give this strong conclusion.
11) The authors should deposite the genomic data in publicly shared repositories or link to access the data in the papar
Minor
Table numbering incorrect, please amend it
The authors should include the human ethics approved number in the manuscript and indicate how informed consent is obtained.
Reviewer 3 Report
This study is to investigate the association of FGFR-2 mutation with β-catenin, which might provide prognosis information for endometrial cancer patients. Here are the major concerns about this study:
1. The authors claim that FGFR-2 mutation worsen the prognosis of endometrial cancer. However, there is no survival or disease progression analysis of this correlation to demonstrate that.
2. The authors claim that FGFR-2 mutation worsen the prognosis of endometrial cancer via epithelial-mesenchymal transition, but there is no mechanistic analysis or functional experiment to demonstrate that.
3. According to the Materials and Methods, there are many assays as well as sequencing performed in this study. However, only minimal data was shown in the results, which makes it hard to be interpreted. A better illustration of the results should be shown.
4. An extensive English editing is required to make the article readable.
Round 2
Reviewer 1 Report
The authors have made improvements to the overall study including additional experiments to support their claims and clarification of experimental design and methods.
Author Response

(The authors gave the same response as above.)

Reviewer 2 Report
Thank you for submitting the revised version of the manuscript and for the efforts have been made to improve the manuscript. The authors partially responded the inquiries raised in the first review, but still revised version is not adequate.
1. The major concern is the data presented does not support the title of the article and the claimed conclusion. Title "FGFR-2 Mutation Worsens Progno-sis Endometrial Cancer via Epithelial–Mesenchymal Transition". Conclusion" the EMT, could in great part be caused by FGFR-2 amplification".
In the whole manuscript there is no any single data that represented the association of FGFR2 mutation with grade, stage, EMT markers and outcome (over all survival (OS), Endometrial cancer specific survival (ECSS), recurrence free survival (RFS)).
It is well documented FGFR2 mutation is associated with survival outcome in endometrial cancer. The authors did not referenced the two big articles that first published the association of FGFR2 and progression in EC in this manuscript which deserve to be included.
To show the association of FGFR2 mutation with progression and EMT markers, the authors need to present the data in this table format
Variables FGFR2 mutant FGFR2wild-type P-value
Age (category
BMI (BMI category)
Histologic type
*Endometrioid
Non-endometrioid
LVSI+
LVSI-
Vimentin high
Vementin low
E-cad high
E-cad low
Tumour budding high
Tumour budding low
#FIGO grade
Grade 1
Grade 2
Grade3
FIGO Stage
Stage I/II
Stage III/Iv
Survival outcome
OS
ECSS
RFS
* adenosquamous carcinoma is and an endometrioid carcinoma and please include as one entity with endometrioid
# The standard grading system in practice for endometrial cancer is FIGO grading
# the standard staging in practice for endometrial cancer is FIGO staging and authors should follow this grading and staging
2.Provide Kaplan Miere Curve (KMC) using Log-rank test probability that show FGFR2 mut vs FGFR2 wildtype.
3.To conclude FGFR2 mutation is associated with progression in endometrial cancer the authors should show FGFR2 mutation is independent prognostic in univariable and multivariable Cox regression proportional hazard model.
In this data serous endometrial cancer histologic type is not mentioned or presented. Were they excluded from study or not in this cohort of patients? But 20% P53 mutation is reported and p53 mutation is common in serous type or non-endometrioid histology. Justification is needed in this presentation.
There are several in appropriately presented ideas throughout the manuscript that needs to be addressed.
In conclusion "the EMT, could in great part be caused by FGFR-2 amplification" This statement is not supported by data. FGFR2 mutation is not same as FGFR2 amplification. FGFR2 amplification very rare in endometrial cancer. The authors should explain what message want to conclude.
"FGFR inter-acts directly with cadherins (and NCAMs) through the conserved acid-box region in IgIII [19-21].26–29] and then activates many pathways. In the present study on β-catenin we observed its nuclear expression or focal loss of membrane expression in APC, CTNNB-1, or CDH-1 negative tumors. Moreover, these tumors were FGFR-2 positive. The results concerning its impact on OS, grade, and stage were positive, but this may be an additional result of FGFR activation. In our study, we presentedfound that the loss of tumor cell-scells’ integrality by tumor budding and together with vimentin expression it vividly pre-sentspromotes the EMT. We observed a tight correlation between the EMT with progress of and advanced grade and stage and with general OS". No data that show this concept at all. What does mean FGFR2+.
The authors need to understand EMT is a process not a single biomarker. Please correct the concept.
I strongly believe that the manuscript will improved by language proficient expert review. If reviewed provide the certificate of language review.
The authors mentioned data pertaining to Mismatch repair proteins (MMR) status in a separate paper. I highly recommended to be disclosed this paper/ manuscript to reviewer to exclude overlapping data in both articles.
Ethics concern. This study is retrospective hospital based data. How did patients get consented retrospectively. Please give justification. I believe waver of consent could be obtained for retrospective data analyses.
Reviewer 3 Report
The authors have improved the manuscript significantly but there are still some comments that I would like to point out:
1. Again, the title of the manuscript is “FGFR-2 mutation worsens prognosis of Endometrial Cancer via Epithelial-Mesenchymal Transition”. The authors took a long way to demonstrate that. The authors showed that there is a weak correlation (R=0.2508) between FGFR-2 mutation and beta-catenin expression, and beta-catenin expression has a weak correlation (0.2049) with the tumor progression. Why don’t the authors just show the correlation of FGFR-2 mutation with the tumor progression or the survival to support the claim in the title? Otherwise, the authors might want to reconsider changing the title into something that can be supported by the data such as LVI or vimentin expression.
2. This leads to the second comment that I have about the interpretation of multiple variable logistic regression analysis that the authors used extensively in this study. Here is a good article about the correlational analysis which is widely cited in the Medicine field from Chan Y. H., entitled “Biostatistics 104: correlational analysis” (PMID: 14770254). Please consult a Biostatistician to confirm the interpretation of the multivariable logistic regression. If I understand correctly, according to Chan, the strength of a linear relationship is considered “weak” or “poor” if the correlation coefficient value is lower than 0.3. If that is the case, the authors might need to revise the interpretation of the data.
3. The English in the article has been improved significantly. However, there are still many errors possibly due to direct translation. For example, gene “battery” on page 4 which is likely to be gene “panel”; “confrontation on page 8 which is likely to be “comparison”; “join” on page 8 which is likely to be “correlate”; and etc… It would be better to have a scientific editor to further improve the article professionally.
